# Effect of Single and Two-Cycles of High Hydrostatic Pressure Treatment on the Safety and Quality of Chicken Burgers

**DOI:** 10.3390/foods12203820

**Published:** 2023-10-18

**Authors:** María Luisa Timón, Irene Palacios, Montaña López-Parra, Jonathan Delgado-Adámez, Rosario Ramírez

**Affiliations:** 1Food Technology, Agriculture Engineering School, University of Extremadura, Avda. Adolfo Suárez s/n, 06007 Badajoz, Spain; mltimon@unex.es; 2Technological Institute of Food and Agriculture (INTAEX), Centro de Investigaciones Científicas y Tecnológicas de Extremadura (CICYTEX), Avda. Adolfo Suárez s/n, 06007 Badajoz, Spain; irenepalaciosromero@gmail.com (I.P.); montana.lopez@juntaex.es (M.L.-P.); jonathan.delgado@juntaex.es (J.D.-A.)

**Keywords:** single and two cycles of high hydrostatic pressure, chicken burger, safety, color, oxidative status, sensory analysis

## Abstract

The aim of this study was to evaluate the effect of two cycles of high hydrostatic pressure (HHP) treatment on chicken burgers after storage at refrigeration (4 °C) for 15 days, in comparison with the application of a single cycle of high hydrostatic pressure treatment, as well as compared with non-treated burgers. Samples were treated at 400 and 600 MPa and a single or two cycles were applied. The results showed that mesophilic, psychrotrophic molds, yeast, and coliforms were significantly reduced by HHP treatment (*p* < 0.05), 600 MPa/1 s (2 cycles) leading to the maximum inactivation. Concerning color parameters, a significant increase in lightness/paleness (L*) and a reduction in redness (a*) and yellowness (b*) (*p* < 0.05) was observed in samples as 600 MPa were applied. Moreover, 600 MPa/1 s (2 cycles) caused the highest differences in the meat color (*ΔE* processing) of the chicken burgers. No HHP treatment significantly affected the degree of oxidation of samples (*p* > 0.05). However, 600 MPa/1 s (2 cycles) samples showed the highest values of TBA RS content after 15 days of storage (*p* < 0.05). Finally, the appearance, odor, taste, and global perception of cooked burgers were similar in all groups (*p* < 0.05). Therefore, treatments at 600 MPa produced a significant reduction in microbial counts but modified the color; however, the discoloration effect in the cooked burgers was not noticed by panelists.

## 1. Introduction

Due to population growth, urbanization, and rising incomes in developed countries, poultry meat consumption has increased considerably in recent decades [1]. Chicken meat is rich in protein, containing less fat than most beef and pork cuts. Furthermore, this meat is a good source of phosphorous and other minerals, and vitamins of B-complex [2]. Within this context, ground chicken meat has gained considerable attention in recent years as a lean and versatile alternative to ground beef or read meats [3].

However, poultry meat consumption raises serious food safety concerns, as they are the source of numerous food poisonings. In this regard, poultry products, such as chicken meat, are among the most frequently implicated in foodborne outbreaks worldwide [4]. Chicken meat can be contaminated with a large number of microorganisms, including those able to spoil the product when it is stored under refrigeration, and some foodborne pathogens. These include bacteria such *Escherichia coli*, *Staphylococcus aureus*, *Salmonella* spp., and *Listeria monocytogenes*, the main food contaminants worldwide and the major causes of gastroenteritis in humans [5]. Many studies revealed that a significant percentage of ground chicken samples were positive for *Salmonella*, *Escherichia coli*, and *Listeria monocytogenes* [6]. In this sense, the U.S. Department of Agriculture’s Food Safety and Inspection Service (FSIS) reported that, despite there being a clear reduction in *Salmonella* in chicken products, 7% of chicken tested positive for *Salmonella* from July 2021 to June 2022, leading to the possibility of illness if a raw product is improperly cooked [7]. In addition, the microbiological spoilage contamination of fresh poultry products causes serious economic losses and consequences on human health [8], which makes it necessary to develop practical methods to reduce this microbiological contamination.

Due to lifestyle changes in recent decades, food consumption patterns have been significantly modified with an increased demand for fresh and minimally processed foods. Likewise, consumers are increasingly concerned about the quality and microbiological safety of fresh products, and they demand healthier and safer ones, and ones with a fresh appearance. This all explains the importance of implementing techniques such as hydrostatic high pressure (HHP), a non-thermal food processing technique that has gained significant attention in recent years due to its ability to inactivate pathogens and spoilage microorganisms while preserving the quality attributes of food [9,10,11,12]. The safety considerations and regulatory aspects associated with HHP-treated foods have also been widely described [13].

HHP consists in the application of pressures from 400 MPa up to 600 MPa, causing critical damage to the microorganisms present in food [14,15]. For microbial inactivation, the HHP treatment required depends on the type of microorganism, the growth stage, and the level of pressure applied as well as the matrix composition. HHP treatment, at low or moderate temperatures, results in microbial vegetative cell inactivation, but it is not effective for spore deactivation, which show high pressure resistance and require higher pressure treatments (>1000 MPa) [16]. For the inactivation of bacterial and fungal spores, multi-pulse HHP treatment effectiveness has been documented [17]. This multi-pulse treatment consists of repeated cycles of compression, holding time, and decompression, which could reduce the pressure needed to inactivate microorganisms [14,17]. Furukawa et al. [18] indicated that repeated cycles were effective to inactivate bacterial spores, since spore germination would be initiated during the first pressurization and the resulting vegetative cells would be increasingly injured by the second pressurization.

Regarding this, HHP technology holds great promise as an alternative meat processing technique that can meet the increasing demands for safe, minimally processed, and high-quality meat products. The applications of HHP in meat, focusing on its effects on microbial inactivation, enzymatic activity, shelf life extension, texture modification, and sensory attributes and the factors influencing the efficiency of HHP treatment, process parameters, and challenges associated with its implementation have been discussed in beef patties [19]; caiman meat [20]; cured meat sausages [21,22,23]; goose meat [24]; turkey meat [25]; dry cured meat [26]; or chicken meat [27,28,29]. In this sense, Cap et al. [27] reported that HHP reduced *Salmonella* spp. in frozen chicken breast when 400 MPa for 5 min and 500 MPa for 1 min were applied. However, significant modifications were found in color and texture parameters compared with control samples. In another study about marinated ground chicken breast [28], a treatment at 600 MPa for 15 min significantly reduced the total aerobic mesophilic bacteria and total mold and yeast. Moreover, the total *Salmonella* spp. and total coliform counts were also reduced to below detectable levels. Nevertheless, modifications affecting physical, chemical, and textural properties were not adversely affected with the treatment and the total antioxidant capacity, and the total phenolic, color, and sensory properties were even improved. Specifically, color acceptance significantly increased with the HHP treatment after cooking, whereas no significant changes were observed in aroma, hardness, and aftertaste. In this respect, Kruk et al. [30] described an improvement in color and aroma acceptance in soy sauce/olive oil chicken breast meat treated by HHP (300 or 600 MPa). On the contrary, Kruk et al. [31] found that increased pressure (300, 450 and 600 MPa) had a significant negative effect on the flavor, aroma, hardness, and juiciness of chicken breast fillet after cooking.

On the other hand, the effect of the combination of HHP treatment with the use of antimicrobial compounds or tenderizers has been described in others research about chicken meat, showing a higher microorganism inactivation without affecting color very significantly [29,32,33].

Despite the fact that the effectiveness of multi pulse high hydrostatic pressure treatment on bacterial inactivation has been demonstrated [17], to our knowledge, the application of this technology to food has not been investigated yet. Therefore, the aim of this study was to evaluate the effect of two cycles of high hydrostatic pressure treatment on the safety and shelf-life of chicken burgers after storage at refrigeration (5 °C) for 15 days, in comparison with the application of a single cycle of high hydrostatic pressure treatment.

## 2. Materials and Methods

### 2.1. Raw Material and Chemicals and Burgers Manufacture

Chicken burgers were prepared with vacuum-sealed, frozen chicken breast (acquired in a local market, grupo SR). Tempered meat (2°) was finely chopped (Mainca mincing equipment) and seasoned in the following proportions. For each kg of minced meat, 0.5 g of black pepper was added; 0.75 g of white pepper; 1.5 g of chopped dehydrated onion; 1.5 g of dehydrated parsley; and 13 g of salt. This proportion has been optimized thanks to previous studies carried out. After mixing meat, salt, and spices (Mainca vacuum mixing unit), burgers of 100 g were prepared using round molds adequate for burger preparation. Burgers were individually vacuum-packaged in 20 × 20 cm plastic bags (OptiDure™ ODA7005 plastic bags, (oxygen permeability: 10 cm^3^ m^−2^, 24 h^−1^, and 0% relative humidity); Cryovac, Madrid, Spain) for the HHP treatment. Vacuum packaging (−0.8 bar) was performed using Henkovac Proeco equipment (Henkovac International, Hertogenbosch, The Netherlands).

### 2.2. Hydrostatic High Pressure Treatment

Vacuum-packaged burgers were pressurized in a semi-industrial hydrostatic pressure unit with 55 L of capacity (Hiperbaric Wave 6000/55; Burgos, Spain). The unit is located in the pilot plant of our Institute. Initial water temperature inside the vessel was 10 °C. Measuring the extent of the adiabatic heating directly in the pressure unit was not possible; it was calculated theoretically according to the Equation (1):dT/dP = α × T/ρ × Cp(1)
where T is temperature (K), P is pressure (Pa), ρ is density (kg/m^3^), Cp is heat capacity of the food substance at a constant pressure (J/kg K), and α is thermal expansivity (K^−1^). The temperature increase due to adiabatic heating was approximately 2.5 °C/100 MPa. Therefore, temperature within the pressure vessel increased from 10 to 20, and 25 °C for treatments at 400, and 600 MPa, respectively. Time to reach final pressure (400 or 600 MPa) was about 4 min. Decompression of the vessel was instantaneous (<3 s). After processing, packages were stored at refrigeration (4 °C) for 15 days in darkness.

### 2.3. Experimental Design

In total, 50 burgers were manufactured (5 samples per batch × 5 treatments (1 control + 4 HHP conditions) × 2 times of storage), and 5 additional control burgers were prepared for the proximate composition analyses. The burgers, once packed in vacuum bags, were subjected to high hydrostatic pressure treatments between 400 and 600 MPa. These two pressure intensities were chosen because of their effect on microbial inactivation (400–600 MPa). At the commercial level, 600 MPa is the maximum intensity of pressure generally applied. Short holding times were applied because the effect of pressure intensity is generally higher than holding time. In some cases, two consecutive processing cycles were applied to enhance microbial inactivation. In the first cycle, sublethal damage occurs, making these microorganisms more sensitive in a second cycle. The processing conditions are detailed below: HHP1: 400 MPa/1 s; HHP2: 400 MPa/1 s (2 cycles); HHP3: 600 MPa/1 s; HHP4: 600 MPa/1 s (2 cycles). Time between the first and the second cycle was 20 h in both HHP2 and HHP4. After processing, burgers were stored at refrigeration (4 °C) for 15 days. Burgers were sampled on day 1 (day after processing) and after 15 days of refrigerated storage. This sampling time was chosen to compare the effect of the different HHP conditions applied and compared with control burgers.

### 2.4. pH, Composition Analyses and Fatty Acid Determination

The pH determination was carried out using a puncture pH-meter specific for meat products (HANNA, model HI 99163, Instrumentación Científica y Técnica, S.L, La Rioja, Spain). Moisture content of burgers was determined following the ISO recommended method [34]. Total lipids of samples were extracted according to the method described by Folch, Lees, and Stanley [35] using trichloromethane: methanol (2:1 *v*/*v*). Protein content (Kjeldahl N × 6.25) was determined following the official method [36]. Fatty acid profile was determined following the method used by Ortiz et al. [37] with KOH in methanol and analyzed by GC-FID. One measurement per sample (burger) was performed.

### 2.5. Microbiological Analysis

Ten grams of each sample were taken aseptically and homogenized with 90 mL of peptone water (Merck, Darmstadt, Germany) in a laboratory blender (Stomacher^®^ 400 Circulator). Serial decimal dilutions were made in sterile peptone water and 1 mL samples of appropriate dilutions were poured or spread onto total count and selective agar plates. Except for *S. aureus*, 0.1 mL of the appropriate dilution was applied to the selective medium for surface seeding.

Total aerobic mesophilic bacteria and psychrotrophic counts were performed using a standard Plate Count Agar (PCA) (Merck, Darmstadt, Germany), which were incubated at 30 °C for 72 h and 6.5 °C for 10 days, respectively. From appropriate 10-fold dilutions, molds and yeasts were enumerated on Yeast Extract Glucose Chloramphenicol Agar (Merk, Darmstadt, Germany) incubated at 25 °C for 5 days. Total coliforms and *Escherichia coli* were incubated on Chromocult Agar (Merck, Darmstadt, Germany) at 37 °C for 24–48 h; *Staphylococcus aureus* were determined on Baird Parker Agar (Merck, Darmstadt, Germany) after incubation at 37 °C for 24–48 h; *Clostridium perfringens* were incubated on Triptosa-Cicloserina-Sulfito Agar (Merck, Darmstadt, Germany) at 37 °C for 24 h under anaerobic conditions. Results were expressed as Log_10_ CFU (colony forming units) g^−1^. The detection limit of the above techniques was 10 CFU g^−1^, except for *S. aureus* which was 100 CFU g^−1^.

Finally, *Salmonella* spp. and *Listeria monocytogenes* were determined according to ISO 6579, 1993 [38] and ISO 11290-1, 2017 [39], respectively. The results were expressed as the absence or the presence of *Salmonella* spp. or *L. monocytogenes* in 25 g of burgers.

### 2.6. Instrumental Color Parameters

The color parameters were reported in relation to the CIELAB scale (L*, a*, and b*), where L* is related to luminosity, a* refers to redness (+a*) or greenness (−a*), and b* indicates yellowness (+b*) or blueness (−b*). Color measurements were performed using a Konica Minolta CM-5 Spectrophotometer (Konica Minolta, Tokyo, Japan). Reflectance was measured using an illuminant D65 with a viewing angle of 10°. The diameter of the aperture was 30 mm. Hue angle, which describes the hue or color (h°), was calculated (h° = arctan − 1 (b*/a*)) as well as the saturation index or Chroma (C*) (C = (a*^2^ + b*^2^)^0.5^) which describes saturation of color.

Total color differences (*Δ*E) were calculated to evaluate the effect of (i) the processing conditions of burgers (*ΔE*processing: control vs. high pressure-treated burgers) using the Equation (2); and (ii) the changes during the storage (*ΔE*storage: day 1 vs. day 15) using Equation (3):(2)$${\Delta E}_{processing}=\sqrt{{\left(L^{*}-L_{0}^{*}\right)}^{2}+{\left(a^{*}-a_{0}^{*}\right)}^{2}+{\left(b^{*}-b_{0}^{*}\right)}^{2}}$$
(3)$${\Delta E}_{storage\;(day\;1-day\;15)}=\sqrt{{\left(L_{2}^{*}-L_{1}^{*}\right)}^{2}+{\left(a_{2}^{*}-a_{1}^{*}\right)}^{2}+{\left(b_{2}^{*}-b_{1}^{*}\right)}^{2}}$$

L*_0_, a*_0_, b*_0_ are the values control burgers (non-treated); L*, a*, b* are the values of burgers after HHP. L*_1_, a*_1_, b*_1_ are the values for burgers at day 1 and L*_2_, a*_2_, b*_2_ are the values at day 15 of storage with the same processing conditions, to evaluate changes after storage period. One measurement was made on each side of the non-packaged burgers and averaged.

### 2.7. Oxidative Status

Lipid oxidation was assessed using the thiobarbituric acid reactive substances (TBA-RS) method [40], using a standard curve of tetraethoxypropane.

Protein oxidation was evaluated by measuring the carbonyl groups formed during incubation with 2,4-dinitrophenylhydrazine (DNPH) in 2 N HCl [41].

### 2.8. Sensory Analysis

For the sensory analysis, an independent assay was replicated with 5 burgers per batch (5 × 5 treatments/control). Only burgers at day 1 were tasted because of panelist’s safety. Cooking conditions were previously optimized (ie. optimal internal temperature, cooking time, cooking devices, etc.) and the internal temperature of the burger was controlled (72 °C). Each burger was cooked in a contact grill (Lacor, 43 × 35 × 19 cm; 1400 w) at maximum temperature (220 °C) for two minutes. Each burger was cut into four pieces after cooking and wrapped in aluminum foil. A random identification number was used for each batch. Mineral water and bread were provided to each panelist. One sample per batch was tasted by each panelist, so a total of 5 pieces were tasted by each person. Twenty non-trained panelists participated in the sensory analysis with ages between 20 and 65 years. The following parameters were evaluated in a 10 cm-non-structured scale (from I dislike all to I like very much): general appearance, odour, taste, texture, global acceptability. In addition, the presence–absence of undesirable tastes was analyzed (absence–presence).

Informed consent was obtained from all subjects involved in the study. The study was conducted in accordance with the Declaration of Helsinki and approved by the Institutional Review Board (or Ethics Committee) of University of Extremadura (111/2002, 16 June 2022), for studies involving humans. This research was funded by CRUCIFOOD project (co-funded by European Regional Development Fund (ERDF) Operational Program for Extremadura 2014–2020).

### 2.9. Statistical Analysis

Data were analyzed using the statistical software package SPSS v.21.0 (IBM-SPSS, Chicago, IL, USA). Descriptive analysis was performed in order to calculate means and standard error of the mean of the measurements of each parameter. Data were analyzed using a one-way analysis of variance applied twice to know the effect of HHP treatment and storage as main effects. Significance was defined as *p* < 0.05. In cases where the effect of some of these independent variables was significant, the means were compared using the Tukey test (*p* < 0.05). In addition a two-way ANOVA was carried out in order to know the effects of processing and storage and their interaction.

## 3. Results and Discussion

### 3.1. Proximate Composition, pH, and Fatty Acid Profile

The values of the pH, total fat, moisture, protein, and fatty acid profiles of chicken burger are shown in Table 1. The pH values of manufactured chicken burgers (5.96) were consistent with those provided by other authors in chicken breast [28,32]. The moisture and protein content (73.97% and 20.87%, respectively) were similar to the values described by other authors in breast of different chicken varieties [42,43]. However, the content of total fat in our sample (1.82%) were different to those presented by these same authors. In this sense, Hailemariam et al. [42] reported that fat content may vary widely in meat chicken depending on genotypes, sexes, ages, husbandry, and diet.

Related to the fatty acid profile, a total of nine fatty acids, including three saturated (SFA) (C14:0, C16:0, C18:0), four monounsaturated (MUFA) (C16:1, C17:1, C18:1, C20:1), and two polyunsaturated (PUFA) (C18:2, C18:3) were identified. The content of SFA, MUFA, and PUFA were around 32%, 41%, and 25%, respectively, which were very different to the values found by Li et al. [44] and Lian et al. [45]. According to these authors, age, feed, and the varieties of chicken are important factors affecting fatty acid profiles in chicken meat; therefore, different factors among studies may considerably modify fatty acid profiles.

### 3.2. Microbiological Analysis

Microbiological counts of HHP treated chicken burger before and after refrigerated storage are presented in Table 2. Before storage, mesophiles were significantly reduced by all HHP treatments (*p* < 0.05). The levels of mesophiles were below the detection limit (<1 log CFU g^−1^) when 600 MPa/1 s (two cycles) (HHP4) was applied to the samples; however, the mesophilic count in 400 MPa/1 s (two cycles) (HHP2) was higher than the count in 400 MPa/1 s (HHP1). Regarding psychrotrophic microorganisms, the same results were obtained, with HHP4 leading to maximum inactivation (<1 log CFU g^−1^) (*p* < 0.05); however, HHP2 and HHP3 did not affect this type of microorganism (*p* > 0.05). On the other hand, all HHP treatment reduced mold and yeast populations below the detection level (<1 log CFU g^−1^) (*p* < 0.05), the same as treatments HHP2, HHP3, and HHP4 with total coliforms (*p* < 0.05). Finally, counts of *E. coli*, *S. aureus* and *Cl. perfringens* were always below the detection limit in all batches, including the control samples. *Salmonella* and *Listeria* were also non-detected. These results prove the effectiveness of HHP treatment to reduce microbiological counts, with mesophilic and psychrotrophic populations being more affected as two cycle treatments at 600 MPa/1 s were applied, causing the highest log reduction. In this sense, Evrendilek [28] reported that treatments between 450 and 600 MPa provided the complete inactivation of mesophilic, total salmonellae, coliforms, molds, and yeasts in marinated ground chicken breast, with 600 MPa providing the highest reduction (6–8 log). As it is demonstrated, HHP treatments cause structural changes in microorganism cell membranes affecting membrane permeability and cell stabilization resulting in cell death [46]. The increase in permeability of the membranes creates sublethal damage that, during storage, can be remedied by the bacteria’s access to nutrients, but a second cycle would destroy these populations as they are in a disadvantaged situation and therefore greater effectiveness would be expected. The reduction in microbiological counts by HHP treatment has also been described in other meat products, such as caiman meat, where 200 MPa, 300 MPa, and 400 MPa caused lower loads of psychrotrophic, lactic acid, and mesophilic bacteria [20]; in dry-cured meat products, mesophilic, mold, and yeast populations were significantly reduced using 600 MPa for 8 min [26]. However, 600 MPA for 7 min did not reduce the counts of mesophilic, psychrophilic, lactic acid bacteria, total coliforms, mold, and yeast in Iberian dry cured ham [47]. This low reduction in microbial population could be associated with the low water activity and the initial microbial counts of the products.

After 15 days of refrigerated storage of the chicken burgers, mesophilic counts were significantly increased in all the treated and non-treated samples (*p* < 0.05). A significant interaction was also found among the batches, with HHP4 showing a lower count than the other batches (*p* < 0.05). No significant differences occurred when treatment was applied at 400 MPa/1 s, regardless of undergoing one or two cycles. Psychrotrophic counts also increased after 15 days of storage, excepting in the HHP3 and HHP4 samples (*p* > 0.05). Interaction treatment-storage showed that 600 MPa/1 s (one or two cycles) led to the lowest psychrotrophic load. Yeasts and molds and total coliform counts only increased in the control and HHP1 samples after 15 days of storage. When the interaction was studied, the non-treated samples (control) presented the highest values of these microorganisms. On the other hand, *E. coli*, *S. aureus*, and *Cl. perfringens* were below the detection limit in all the batches after 15 days of refrigerated storage, excepting *S. aureus* in the control samples. Therefore, it is evident that HHP treatment is effective on the safety of chicken burger after 15 days of refrigerated storage, specifically when HHP4 (600 MPa/1 s (two cycles)) treatment was applied. In relation to this, Canto et al. [20] also described an increase in psychrotrophic, lactic acid, and mesophilic bacteria after vacuum storage at 4 °C in caiman meat treated with HHP; however, treatment at 400 MPa showed lower bacterial loads in comparison with 200 and 300 MPa. Omer et al. [21] also reported higher counts of mesophilic, psychrotrophic, and lactic acid bacteria after ripening in dry fermented sausages manufactured with raw meat treated at 600 MPa. However, the *Enterobacteriaceae* count was reduced to a no-detection level in these samples after the storage. According to European regulations (Commission Regulation (EC) No 2073/2005) [48], after 15 days of storage, only burgers treated at 600 MPa (HHP3 and HHP4) present adequate counts for consumption, although the HHP would be at the limit.

### 3.3. Color Parameters

The effect of different HHP during refrigerated storage on the chromatic parameters and total color differences of chicken burgers are shown in Table 3 and Table 4.

Among the parameters presented in Table 3, it is particularly noteworthy how excessively low the values of a*, ranging from 2.7± 0.6 to −0.7 ± 0.6, are compared with other studies in chicken meat [29,32,33]. The ingredients and spices added in meat burger manufacturing could cause this a decrease in this parameter, since some ingredients, like parsley with green color, were visible on the surface of the burger. In this regard, Ros-Polski et al. [29] described a loss of redness in added NaCl ground chicken meat.

Concerning L* values before storage (Table 3), HHP4 samples exhibited a significantly paler color (L = 73.3 ± 1.0) than the rest of the samples (control or HHP treated) (*p* < 0.05), with L* values being higher as the pressure increased (*p* < 0.05). In this line, previous studies have reported that the application of HHP treatment led to a “whitening” effect on chicken meat [27,28,32,33]. In this sense, Ros-Polski et al. [29] suggested that myoglobin denaturation and the coagulation of sarcoplasmic and myofibrillar proteins could cause an increase in the lightness of HHP treated meats.

Regarding the a* values, HHP3 and HHP4 showed the lowest values (*p* < 0.05). The loss of redness has also been described by other authors who reported that the appearance of ground chicken meat became less red when HHP treatments were applied, with this observation likely being due to the oxidation of globin hemochrome (ferrohemochrome) to brown ferrihemochrome reaction (globin hemichrome) [33]. Similar results were found in pork slurries [49] and in pork sausages [50]. On the contrary, Chen et al. [32] reported an increase in a* values in raw chicken breast with pressures up to 450 MPa. Ros-Polski et al. [29] did not observe any trend for this parameter in chicken breast using pressures up to 600 MPa, the same as Evrendilek [28] in marinated ground chicken breast at 200–600 MPa. According to Ros-Polski et al. [29], the different redness patterns in meat due to HHP treatment could be explained by the redox chemistry of myoglobin, with different pressures causing the activation or deactivation of enzymes that reduce metmyoglobin (brown pigment) to oxy or deoxymioglobin (red pigments).

CIE b* values also decreased with the application of HHP, HHP3 being the treatment that produced the lowest values of this parameter (*p* < 0.05). In this sense, Zhang et al. [24] also described a decrease in the b* values of goose meat after application of HHP. However, other studies related to chicken meat treated with HHP showed an increase in b* values [27,28,32]. In relation to this, Chen et al. [32] also found conflicting results about b*, indicating that this parameter is not always affected by HHP.

Concerning the C* and h* parameters, the values ranged from −11.2 ± 1.2 to 6.8 ± 1.0 and from 13.5 ± 06 to 17.5 ± 0.6, respectively, which were much lower than the ones reported in other studies about ground chicken meat color [51]. These results could be a consequence of the low a* values obtained in this work. The effect of HHP on these parameters is demonstrated, with the highest C* (intensity) and the lowest h* (hue) values being shown as the pressure conditions increased until 600 MPa (HHP3 and HHP4) (*p* < 0.05). Related to the C* parameter, similar results were found by Cap et al. [27] who reported greater intensity (>C* values) in chicken breast fillets when 400 and 500 MPa were applied. On the contrary, Janardhanan et al. [52] did not describe any effect of HHP on the h* parameter in veal patties.

After refrigerated storage over 15 days, in an HHP-treated chicken burger, the values of L* and C* remained constant in all batches (*p* > 0.05). The interaction effect of HHP treatment and storage time on these parameters revealed that 600 MPa/1 s (1 or 2 cycles) led to the highest values of L* and the lowest of C*. On the other hand, the a*, b*, and h* values increased in all the treated samples after storage, except in the HHP4 samples (*p* > 0.05). A significant interaction between treatment and storage indicated that a*, b*, and h* presented the lowest values as the highest pressure was applied (600 MPa/1 s (1 or 2 cycles)) after 15 days of storage.

In relation to this, the determination of total color changes (*ΔE* storage) showed significant differences among control or treated samples after storage (*p* < 0.05) (Table 4). *ΔE* was higher in control burgers than in treated burgers, which indicates that changes during storage in the control burgers were higher than those treated. Thus, HHP increased the color stability of burgers during storage. However, the values were lower than 3 in all batches. In this sense, Tananuwong et al. [53] reported that ΔE values lower than 10 are not considered as significant differences in meat color.

On the other hand, the values of *ΔE* processing were higher as the pressure increased (*p* < 0.05), with all HHP treatment except HHP1 showing values above 10. Therefore, it could be stated that HHP2, HHP3, and HHP4 caused a significant difference in the meat color of the chicken burger, with HHP4 treatment causing the highest difference in comparison with the control samples. In this sense, Evrendilek [28] observed that discoloration in ground chicken meat was influenced by the pressure but also by the holding time, which would agree with our results and the increases in ΔE observed between HHP1 vs. HHP2 and HHP3 vs. HHP4. Therefore, the application of two cycles produced increases in color variations (*ΔE*) with respect to the application of one single cycle. The same conclusion was reached by Cap et al. [27], who suggested that discoloration in frozen chicken breast was greater as the time of processing was longer than 1 min.

### 3.4. Lipid and Protein Oxidation

The effect of the different HHP treatments and refrigerated storage on the oxidative status of chicken burger was evaluated (Table 5).

The results showed that the HHP treatments at day 1 of refrigerated storage did not significantly affect the degree of oxidation (lipid or protein) in any sample (*p* > 0.05). After 15 days storage, TBA-RS content increased in all the samples (*p* < 0.05), with HHP4 presenting the highest values (0.5 mg MDA Kg^−1^) (*p* < 0.05). However, this level of MDA was lower than the limit reported by Insausti et al. [54], who indicated a concentration of 5 mg MDA Kg^−1^ meat, as a detectable concentration for consumers. Protein oxidation was not affected by HHP treatments. In addition, the influence of the storage and interaction treatment-storage on this parameter was not observed. Some authors have reported that HHP can induce lipid and protein oxidation in meat products [26,55,56], due to the capacity of peroxides formed to react with both lipids and proteins. In particular, HHP (500 MPa, 10 min, 20 °C) quickened the process of lipid oxidation in commercial chicken patties [57]. Nevertheless, the effect of HHP on lipid and protein oxidation has been scarcely studied, providing conflicting results due to aspects such as the type of meat, pressure treatment, holding time, and storage conditions. In this sense, some studies have shown that the impact of HHP treatment on oxidation processes is generally limited by the antioxidants in the meat and by vacuum packaging [56].

### 3.5. Sensory Analysis

In general, the parameters analyzed in the sensory analysis did not show significant differences among control and treatments (Table 6).

The appearance of cooked burgers, and their odor and taste were similar in all groups (*p* > 0.05). Only the texture tended to be improved after HHP (Table 7), although ANOVA did not show significant differences and Tukey test did so. In addition, non-unpleasant tastes were increased after processing (*p* > 0.05). The global perception of burgers was not modified after HHP (*p* > 0.05). Similar results were found by Evrendilek [28] who detected no significant changes to sensory properties of aroma, hardness, and aftertaste in marinated ground chicken breast. It is likely that the spices added in these studies could provide a strong aroma and taste to all samples, which could have avoided differences among the control and treated burgers. However, another study in chicken breast treated at 300 MPa with some dressing reported an improvement in flavor scores [30], whereas Kruk et al. [31] described a significant negative effect on the flavor, aroma, hardness, and juiciness of unmarinated chicken breast fillet after HHP treatment.

As described above, the texture of burgers was improved after HHP treatment (especially for HHP3). In this sense, recent studies have showed new applications of HHP; this treatment, in combination with K-salts, might be used as a strategy to reduce NaCl in emulsion-type meat products (like chicken sausage), without compromising consumer preference in textural properties [58]. In this respect, Jiménez-Colmenero et al. [59] reported that temperature pressurized chicken batters formed a less rigid gel matrix with more effective water/fat binding properties than non-pressurized ones. Probably, this could explain the better texture perception when chewing the treated chicken burgers with respect to those non-treated in this study.

## 4. Conclusions

It is evident that the effectiveness of HHP treatment on the safety of chicken burger after 15 days of refrigerated storage is increased as the pressure increases. Treatment at 600 MPa/1 s (2 cycles) is more effective in reducing microbiological counts in comparison with 600 MPa/1 s. This effect on microorganisms was not observed at 400 MPa, when a single or two cycles were applied. However, HHP4 treatment (600 MPa/1 s (2 cycles)) caused the highest differences in the meat color of chicken burger, resulting in a meat with a light gray–blue color. However, chicken burger discoloration caused by HHP4 treatment was not perceived by panelists after cooking after one day of refrigerated storage. Therefore, the modifications of color in fresh burger after HHP could probably reduce consumers’ acceptance, but this type of burgers could be very interesting for the HORECA channel in which a safe product with a long shelf-life could be presented.

## Figures and Tables

**Table 1 foods-12-03820-t001:** Proximate composition (g 100 g^−1^), fatty acids profile (%), and pH of the manufactured chicken burger.

	Burgers
pH	5.96 ± 0.01
Moisture	73.97 ± 0.21
Protein	20.87 ± 0.51
Total fat	1.82 ± 0.27
Fatty acids profile	Percentage
C14:0	0.5 ± 0.0
C16:0	24.2 ± 0.5
C16:1	4.6 ± 0.1
C17:1	0.2 ± 0.0
C18:0	8.0 ± 0.4
C18:1	36.5 ± 0.4
C18:2, cis,cis	23.7± 0.3
C18:3	1.9 ± 0.0
C20:1	0.4 ± 0.0

Values are expressed as means ± standard deviation of analyzed samples (*n* = 5).

**Table 2 foods-12-03820-t002:** Microbiological counts (log CFU g^−1^) changes of chicken burgers after high hydrostatic pressure treatment and refrigerated storage.

		HHP Treatment					
	Storage Days	Control	HHP1	HHP2	HHP3	HHP4	*p*-Value
Mesophilic	T0	4.6 ± 0.3 a	2.6 ± 0.4 c	3.9 ± 0.2 b	3.7 ± 0.5 b	<1	0.000
	T1	9.7 ± 0.2 a	8.7 ± 0.1 b	8.7 ± 0.1 b	5.5 ± 0.3 c	3.8 ± 0.2 d	0.000
	*p*-value	0.000	0.000	0.000	0.006	0.000	
Psychrotrophic	T0	3.7 ± 0.2 a	1.7 ± 0.2 b	3.1 ± 0.4 a	3.5 ± 0.1 a	<1	0.000
	T1	8.7 ± 0.2 a	8.8 ± 0.0 a	8.8 ± 0.1 a	3.5 ± 3.1 b	<1	0.000
	*p*-value	0.000	0.000	0.000	0.956	--	
Molds and yeasts	T0	2.7 ± 0.3 a	<1	<1	<1	<1	0.000
	T1	6.5 ± 0.1 a	4.9 ± 0.1 b	<1	<1	<1	0.000
	*p*-value	0.000	0.000	--	--	--	
Total coliforms	T0	2.3 ± 0.3 a	1.1 ± 0.6 b	<1	<1	<1	0.000
	T1	6.7 ± 0.2 a	5.3 ± 0.2 b	1.7 ± 1.5 c	<1	<1	0.000
	*p*-value	0.000	0.000	0.088	--	--	
*E. coli*	T0	<1	<1	<1	<1	<1	--
	T1	<1	<1	<1	<1	<1	--
	*p*-value	--	--	--	--	--	
*S. aureus*	T0	<2	<2	<2	<2	<2	--
	T1	2.9 ± 0.2 a	<2	<2	<2	<2	0.000
	*p*-value	0.021	--	--	--	--	
*Cl. perfringens*	T0	<1	<1	<1	<1	<1	--
	T1	<1	<1	<1	<1	<1	--
	*p*-value	--	--	--	--	--	

Values are expressed as means ± standard deviation of analyzed samples (*n* = 5). *p*-value < 0.05 indicates statistical differences. Different letters (a–d) in the same row indicate statistical differences (Tukey-b’s test, *p* < 0.05). HHP1: 400 MPa/1 s; HHP2: 400 MPa/1 s (2 cycles); HHP3: 600 MPa/1 s; HHP4: 600 MPa/1 s (2 cycles). T0 = initial storage (1 day of refrigerated storage); T1 = 15 days of refrigerated storage.

**Table 3 foods-12-03820-t003:** Instrumental color parameters of chicken burgers after high hydrostatic pressure treatment and evolution during refrigerated storage.

		HHP Treatment					
	Storage Days	Control	HHP1	HHP2	HHP3	HHP4	*p*-Value
CIE L*	T0	55.3 ± 1.2 e	62.9 ± 0.5 d	65.2 ± 0.7 c	71.3 ± 1.5 b	73.3 ± 1.0 a	0.000
	T1	57.4 ± 1.3 c	64.2 ± 0.5 b	65.3 ± 1.1 b	70.9 ± 1.2 a	71.9 ± 0.9 a	0.000
	*P*-value	0.101	0.039	0.896	0.332	0.095	
CIE a*	T0	1.0 ± 0.3 a	−0.3 ± 0.1 b	−0.5 ± 0.3 b	−1.4 ± 0.6 c	−0.7 ± 0.6 bc	0.000
	T1	2.7 ± 0.6 a	0.3 ± 0.3 b	0.1 ± 0.3 bc	−0.5 ± 0.3 cd	−0.7 ± 0.3 d	0.000
	*p*-value	0.006	0.033	0.006	0.014	0.858	
CIE b*	T0	12.2 ± 0.7 a	10.9 ± 0.4 b	10.5 ± 0.4 b	7.8 ± 0.9 c	9.6 ± 1.0 b	0.000
	T1	12.5 ± 0.5 a	11.6 ± 0.5 b	11.4 ± 0.2 b	9.5 ± 0.7 c	9.4 ± 0.5 c	0.000
	*p*-value	0.637	0.032	0.029	0.020	0.613	
CIE C*	T0	−11.2 ± 1.2 e	−3.5 ± 0.5 d	−1.3 ± 0.7 c	4.8 ± 1.5 b	6.8 ± 1.0 a	0.000
	T1	−9.0 ± 1.3 c	−2.3 ± 0.5 b	−1.2 ± 1.1 b	4.4 ± 1.2 a	5.4 ± 0.9 a	0.000
	*p*-value	0.101	0.039	0.894	0.334	0.095	
CIE h*	T0	15.8 ± 0.3 a	14.6 ± 0.1 b	14.4 ± 0.3 b	13.5 ± 0.6 c	14.1 ± 0.6 bc	0.000
	T1	17.5 ± 0.6 a	15.1 ± 0.3 b	14.9 ± 0.3 bc	14.4 ± 0.3 cd	14.2 ± 0.3 d	0.000
	*p*-value	0.006	0.03	0.006	0.014	0.895	

Values are expressed as means ± standard deviation of analyzed samples (*n* = 5). *p*-value < 0.05 indicates statistical differences. Different letters (a–e) in the same row indicate statistical differences (Tukey-b’s test, *p* < 0.05). HHP1: 400 MPa/1 s; HHP2: 400 MPa/1 s (2 cycles); HHP3: 600 MPa/1 s; HHP4: 600 MPa/1 s (2 cycles). T0= initial storage (1 day of refrigerated storage); T1 = 15 days of refrigerated storage.

**Table 4 foods-12-03820-t004:** Instrumental color variation (*ΔE*) of chicken burgers after high hydrostatic pressure treatment and evolution during refrigerated storage.

	HHP Treatment				
	Control	HHP1	HHP2	HHP3	HHP4
*ΔE* processing (control vs. HHP)	--	7.8 ± 0.6 d	10.2 ± 0.6 c	16.8 ± 1.1 b	18.3 ± 0.8 a
*ΔE* storage (Day1-day15)	2.9 ± 1.1 a	1.6 ± 0.5 ab	1.4± 0.5 b	2.3 ± 0.5 ab	1.5 ± 0.9 ab

Values are expressed as means ± standard deviation of analyzed samples (*n* = 5). Different letters (a–d) in the same row indicate statistical differences (Tukey-b’s test, *p* < 0.05).

**Table 5 foods-12-03820-t005:** Changes in lipid (TBA-RS, mg MDA kg^−1^) and protein oxidation (nmol carboxnyls mg protein^−1^) in chicken burgers after high hydrostatic pressure treatment and evolution during refrigerated storage.

	Storage Days	Control	HHP1	HHP2	HHP3	HHP4	*p*-Value
TBA-RS	T0	0.1 ± 0.0	0.1 ± 0.0	0.1 ± 0.0	0.2 ± 0.0	0.1 ± 0.0	0.144
	T1	0.2 ± 0.0 c	0.3 ± 0.1 bc	0.4 ± 0.0 ab	0.4 ± 0.1 ab	0.5 ± 0.1 a	0.000
	*p*-value	0.002	0.001	0.001	0.002	0.003	
Protein oxidation	T0	2.0 ± 0.4	1.9 ± 0.4	1.7 ± 0.2	1.8 ± 0.4	2.1 ± 0.3	0.441
	T1	2.2 ± 0.4	1.8 ± 0.6	1.8 ± 0.6	1.4 ± 0.9	2.5 ± 0.5	0.088
	*p*-value	0.361	0.939	0.858	0.335	0.261	

Values are expressed as means ± standard deviation of analyzed samples (*n* = 5). *p*-value < 0.05 indicates statistical differences. Different letters (a–c) in the same row indicate statistical differences (Tukey-b’s test, *p* < 0.05). HHP1: 400 MPa/1 s; HHP2: 400 MPa/1 s (2 cycles); HHP3: 600 MPa/1 s; HHP4: 600 MPa/1 s (2 cycles). T0= initial storage (1 day of refrigerated storage); T1 = 15 days of refrigerated storage.

**Table 6 foods-12-03820-t006:** Sensory analysis of high pressure treated chicken burgers.

	Control	HHP1	HHP2	HHP3	HHP4	*p*-Value
Appearance (dislike–like very much)	6.1 ± 1.9	6.3 ± 1.7	6.0 ± 2.3	6.6 ± 2.0	6.3 ± 2.2	0.921
Odour (dislike–like very much)	6.3 ± 2.3	6.8 ± 1.5	6.4 ± 2.0	6.3 ± 1.6	6.2 ± 2.1	0.890
Taste (dislike–like very much)	6.5 ± 2.1	6.8 ± 1.9	5.5 ± 2.2	6.8 ± 1.6	6.6 ± 2.2	0.331
Texture (dislike–like very much)	4.6 ± 2.4 b	6.3 ± 1.9 ab	5.3 ± 2.5 ab	6.7 ± 1.5 a	6.0 ± 2.1 ab	0.052
Unpleasant tastes (absence–presence)	1.1 ± 1.7	0.7 ± 1.4	1.4 ± 2.3	1.1 ± 1.6	1.3 ± 2.3	0.891
Global perception (dislike–like very much)	5.9 ± 2.0	6.7 ± 1.8	5.4 ± 2.3	6.7 ± 1.5	6.5 ± 2.2	0.296

Values are expressed as means ± standard deviation of analyzed samples (*n* = 20). *p*-value < 0.05 indicates statistical differences. Different letters (a,b) in the same row indicate statistical differences (Tukey-b’s test, *p* < 0.05). HHP1: 400 MPa/1 s; HHP2: 400 MPa/1 s (2 cycles); HHP3: 600 MPa/1 s; HHP4: 600 MPa/1 s (2 cycles).

**Table 7 foods-12-03820-t007:** Two Way ANOVA. Effect of hydrostatic high-pressure treatment and storage period on preservation of chicken burgers.

Dependent Variable	HHP		Storage		HHP × Storage	
	*p*-Value	Partial Eta Squared	*p*-Value	Partial Eta Squared	*p*-Value	Partial Eta Squared
Mesophilic	0.000	0.972	0.000	0.982	0.000	0.880
Psychrotrophic	0.000	0.850	0.000	0.800	0.000	0.750
Yeast and molds	0.000	0.911	0.000	0.744	0.000	0.745
Total coliform	0.000	0.943	0.000	0.858	0.000	0.832
*S. aureus*	0.000	0.587	0.032	0.110	0.000	0.441
CIE L*	0.000	0.978	0.279	0.029	0.005	0.308
CIE a*	0.000	0.873	0.000	0.507	0.001	0.349
CIE b*	0.000	0.843	0.001	0.256	0.017	0.255
CIE C	0.000	0.975	0.279	0.029	0.004	0.309
CIE h	0.000	0.873	0.000	0.508	0.001	0.350
Δ*E_processing_*	0.000	0.989	0.000	0.613	0.000	0.450
Δ*E_storage_*	0.019	0.250	0.000	0.804	0.019	0.250
TBA-RS	0.000	0.486	0.000	0.839	0.000	0.413
Protein oxidation	0.024	0.240	0.674	0.004	0.498	0.079

## Data Availability

The authors confirm that the data supporting the findings of this study are available within the article and the raw data that support the findings are available from the corresponding author, upon reasonable request.

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
