# Peer review of "Effect of Single and Two-Cycles of High Hydrostatic Pressure Treatment on the Safety and Quality of Chicken Burgers"

_foods, 2023, doi:10.3390/foods12203820_

Round 1

Reviewer 1 Report

The research article, titled “Effect of single and two-cycles of high hydrostatic pressure treatment on the safety and shelf-life of chicken burgers” explored the effect of two cycles of high hydrostatic pressure treatment on the safety and shelf-life of chicken burgers after storage at refrigeration (5 ºC) for 15 days, in comparison to the application of a single cycle of high hydrostatic pressure treatment. The study is within the scope of the journal. The study is almost a routine study with a different product. The data representation is good, and the manuscript is well-written. However, there are concerns as mentioned below:

-The selection of the processing conditions is not clear. It is obvious that increasing the cycle with the same treatment time will increase the effectiveness of HHP. So, why did the authors compare the four mentioned HHP conditions?

-It is better to show the pressure-time profile of the treatments so that the influence of the ‘cycle’ can be visualized. The temperature data should also be mentioned during the HHP cycle.

- It is not clear why the shelf-life study was stopped at 15 days. If the samples were microbially safe for 15 days? If yes, then why it has not been extended to see how long it can be safe? If it was not safe on day 15, then what was the shelf-life of the product? Because there was no intermediate point between day 1 and day 15.

-The authors may apply multivariate analysis to come up with significant conclusions.

-The Introduction section can be minimized significantly. Please see L29-39, 56-84.

L144-145: Why these processing conditions were selected?

L1491: Why the storage time was limited to 15 days? Why there was no in-between points during the shelf-life study?

L205: How is the condition optimized?

L212-214: How these attributes were selected?

Sec 3.1: What was the influence of high-pressure pressurization on the proximate composition of the burger?

Table 2: What is the target log reduction in the product? Based on that which HHP condition is recommended? Please make it clear in Sec 3.2.

L326-329: Please rewrite.

L369: Apart from delta E, the authors may calculate the browning index and compare the data. That might be more realistic for this product.

L399 & 404: Please make it uniform between HHP and HPP.

L392-393: Please discuss this trend with existing literature more critically.

L433-435: Please represent in a different way.

Reviewer 2 Report

The paper by Timón et al. aimed to study the effect of a single and two cycles of high hydrostatic pressure treatment on the safety, shelf-life and sensory acceptance of chicken burgers after refrigerated storage (4 ºC).

The specific comments are as follows:

Abstract

Consider to change: “The aim of this study was to evaluate the effect of two cycles of high hydrostatic pressure (HHP) treatment on the safety, shelf-life and sensory acceptance of chicken burgers after storage at refrigeration (4 ºC) for 15 days, in comparison to the application of a single cycle of high hydrostatic pressure treatment.”. Authors present the Results and Discussion considering also the control (not HPP treated samples), and the sensory analysis was only performed in samples with one day of refrigerated storage.

1. Introduction

 L94-95: Consider to change “…that HHP reduced Salmonella spp in samples when 400 MPa for 5 min and 500 MPa for 1 min were applied in frozen chicken breast. …” to “…that HHP reduced Salmonella spp in frozen chicken breast when 400 MPa for 5 min and 500 MPa for 1 min were applied …”

 L112: Consider to change “…without affecting color very much [32, 29, 33]. “ to “…without affecting color significantly [29, 32, 33].

 2. Materials and Methods

L121: Authors should inform about the origin of the vacuum-sealed frozen chicken breast used for the preparation of the chicken burgers.

L122: Authors should inform about the equipment used for mincing the meat. The temperature was controlled?

L125: Authors should inform about the equipment used for mixing the minced meat with salt and spices.

L139: Consider to change: “Fifty burgers were manufactured (5 samples per batch x 5 HHP treatments/control” to “Fifty burgers were manufactured (5 samples per batch x 4 HHP treatments and control”.

L148: Consider to change:” refrigeration (4 ºC) in for 15 days…” to :” refrigeration (4 ºC) for 15 days…”

L151: How many pH measurements were performed per sample?

L154: Authors should briefly describe the method used for total lipids determination.

L156: Authors should briefly describe the method used for the fatty acid profile determination.

L177: Include reference numbers for ISO 6579 and ISO 11290-1 and in References.

L195: Please clarify how many measurements were performed per sample in each side?

L211-212: Please clarify how were the panelists recruited?

L212: Authors should inform about the length (cm) of the scale used.

3. Results and Discussion

Table 2 – The n value is missing. Authors should include different capital letters to compare values in the same column, in order to better clarify the statistical differences. Please clarify, T0 is time 1 of storage (1 day of refrigerated storage)?

L288-290: “Yeasts and moulds and total coliform counts only increased in control and HHP1 samples after 15 days of storage...”. According to table 2, the control samples present the same letter for these microorganisms in both T0 and T1, please clarify.

L302-304: “…After 15 days of storage, only burgers treated at 600 MPa (HHP3 and HHP4) presented adequate counts for consumption, although HHP would be at the limit.”. Authors should present the reference levels in which they supported the above sentence.

Table 3 – The n value is missing. Authors should include different capital letters to compare values in the same column, in order to better clarify the statistical differences. Please clarify, T0 is time 1 of storage (1 day of refrigerated storage)?

L345: Correct: “13.5±06 to 15.8±0.3…” to “13.5±06 to 17.5±0.6…”

L355: Consider to change “…and storage…” to “…and storage time…”

Table 5 – The n value is missing. Authors should include different capital letters to compare values in the same column, in order to better clarify the statistical differences. Please clarify, T0 is time 1 of storage (1 day of refrigerated storage)?

L391: Please clarify: “The results showed that the HHP treatments before storage…”. Authors are referring to 1 day of refrigerated storage?

Table 6 – The n value is missing.

Authors could have studied the presence of off-odors, performing the sensory analysis of the samples just with one day of storage is limited in terms of shelf-life.

L432-436: Remove the sentence: “This section may be divided by subheadings. It should provide a concise and precise description of the experimental results, their interpretation, as well as the experimental conclusions that can be drawn.”

5. Conclusions

L444-445: “…However, chicken burger discoloration caused by HHP4 treatment was not perceived by panelists after cooking…” This is valid for the samples with just one day of refrigerated storage.

L445-447: This could be interesting to study, but a consumer study is needed using samples stored for longer time.

Overall, the paper is well written.
